# GREEDY DISTILL: EFFICIENT VIDEO GENERATIVE MODELING WITH LINEAR TIME COMPLEXITY

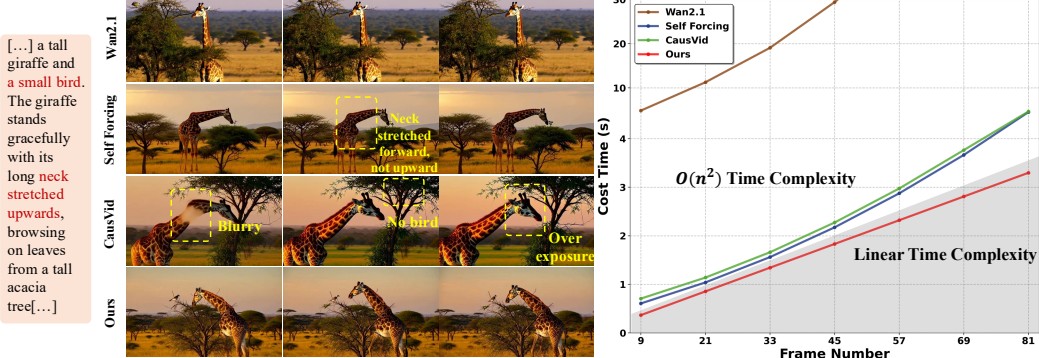

Figure 1: Comparison of results between our Greedy Distill (4 steps), the original Wan2.1 and other distill methods (left). Comparison of inference time across different methods (right) under the video synthesis configuration of 81 frames, which is measured on a single H100 GPU.

## ABSTRACT

Due to bidirectional attention dependencies, video generation models generally suffer from $O(n^2)$ computational complexity. In this work, we find the "local inter-frame information redundancy" phenomenon which indicates strong local temporal dependencies in video generation, with global attention to distant frames contributing only marginally. Built upon this finding, we introduce a novel distillation training paradigm for video diffusion models, namely **GREEDY DISTILL**. Specifically, to generate the next frame using only the 0-th and the last frames, we propose the Streaming Diffusion Decoder (**SDD**) as the "Greedy Decoder" to avoid redundant computational costs from the other frames. Meanwhile, we introduce Efficient Temporal Module (**ETM**) to capture the global temporal information across frames. These two modules achieve the computational complexity reduction from $O(n^2)$ to linear. Moreover, to our knowledge, we make the **first attempt to apply RL fine-tuning** to address the error accumulation during streaming generation. Our method achieves an overall score of 84.60 on the VBench benchmark, surpassing previous state-of-the-art methods by large margins(**+4.18%**). Qualitative results also demonstrate superior performance. Leveraging its efficient model structure and KV cache, it is able to rapidly generate high-quality video streams at **24 FPS** (nearly 50% faster) on a single H100 GPU.

## 1 INTRODUCTION

Video generation models based on diffusion transformers (DiT) (Peebles & Xie, 2023) have achieved remarkable progress. DiT-based video generation models (Ho et al., 2022; Zhang et al.; Blattmann et al., 2023; Hu, 2024; Zheng et al., 2024b; Zhang et al., 2025; Wan et al., 2025) have made significant strides in recent advancements. However, as the demand for long-form video generation grows, the computational cost becomes a critical challenge. The slow iterative sampling process and the reliance on increasingly large denoising networks result in prohibitively high computational requirements, making practical deployment difficult. For DiT-based video generation (Hong et al., 2022; Kong et al., 2024; Yang et al., 2024; Peng et al., 2025), the computational cost is primarily

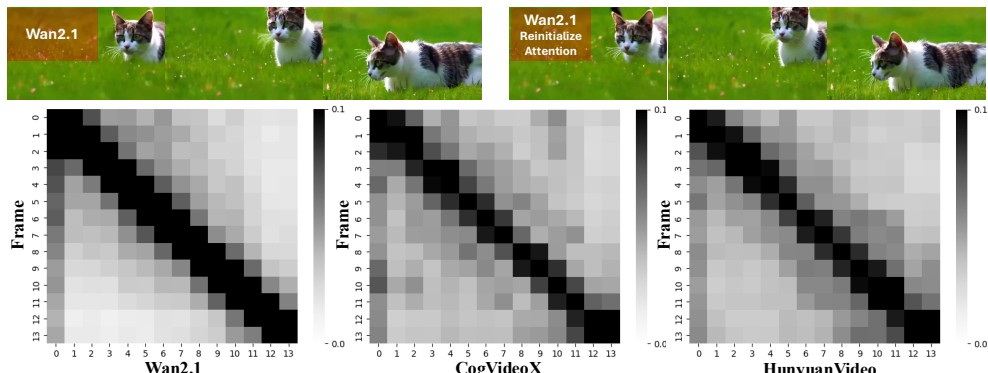

Figure 2: **Local Inter-frame Information Redundancy.** Video generation models assigns substantially higher attention to the first frame and its neighboring frames than to distant ones.

determined by the number of sampling steps ($T$), the number of frames ($F$), and the feature length in the latent space for one chunk of frames ($L$), leading to a total complexity of $T \times F^2 \times L^2$. As a result, generating high-quality videos necessitates substantial computational resources.

To reduce computational overhead, diffusion models for video increasingly adopt a paradigm, where a teacher model guides the training of a student (Luhman & Luhman, 2021; Liu et al., 2022; Salimans & Ho, 2022; Zheng et al., 2023; Meng et al., 2023; Liu et al., 2023). This approach is particularly effective in alleviating the slow, iterative sampling process of video diffusion models. However, existing distillation methods (Yin et al., 2024a;b;c) come with a common trade-off: they focus primarily on reducing computational costs by optimizing the sampling steps, while overlooking other computationally expensive factors. Notably, *optimization along the Frames dimension, which contributes $O(f^2)$ complexity, remains underexplored*. Meanwhile, the recent CausVid (Yin et al., 2024c) employs an asymmetric structural distillation strategy that transfers knowledge from a bidirectional-attention teacher diffusion model to a causal student model. However, *this approach still requires the teacher and student to be nearly isomorphic* and mainly reduces cost by cutting sampling steps.

In our initial exploration, we observe that DiT assigns substantially higher attention to the first frame and its neighboring frames than to distant ones, as shown in Figure. 2; the same phenomenon is evident across multiple video-generation foundation models(*i.e.* Wan (Wan et al., 2025), Hunyuan Video (Kong et al., 2024), CogVideoX (Yang et al., 2024)). Consequently, in these foundation models, we reinitialize the attention matrices as constants whose attention standard variance exceeds 0.02 and observe that the generated videos are nearly indistinguishable from those of the original model. Other work (Meng et al., 2023) reports similar findings; we refer to this as "local inter-frame information redundancy" in video.

The "local inter-frame information redundancy" phenomenon indicates strong local temporal dependencies in video generation, with global attention to distant frames contributing only marginally. Building on this finding, we propose a new asymmetric structural distillation framework (**Greedy Distill**). Specifically, the student model combines an AR Transformer with a diffusion decoder. The AR Transformer adopts a chunk-wise sliding window attention (Berthelot et al., 2023) mechanism as the **E**fficient **T**emporal **M**odule (**ETM**), enabling it to capture strong local and weak global cues to form a temporal representation while avoiding the cost of full global attention. The **S**treaming **D**iffusion **D**ecoder(**SDD**) follows the diffusion paradigm to generate the next frame in a streaming manner, conditioned on the 0-th, the last frame and the temporal representation.

With causal attention and a sliding-window mechanism with window size $w$, the AR Transformer in ETM reduces the $(L^2) \times (F^2)$ term to $(L^2) \times w$, and unlocks streaming generation. The SDD uses the 0-th, the last frames and the temporal features of the ETM to generate the next frame, reducing the original diffusion complexity from $T \times (F^2) \times (L^2)$ to $T \times F \times (L^2)$. The total complexity becomes $(F \times w) \times (L^2) + T \times F \times (L^2) = (w + T) \times F \times (L^2)$, where $L$, $S$ and $w$ are constants, substantially lower than $T \times (F^2) \times (L^2)$. Our approach can also leverage step distillation to further reduce $T$, yielding additional savings. Meanwhile, to address the inevitable error accumulation, we make the first attempt to apply RL fine-tuning to address the exposure bias (Schmidt, 2019; Ning et al., 2023), where a model is trained exclusively on ground-truth context but must rely on its own

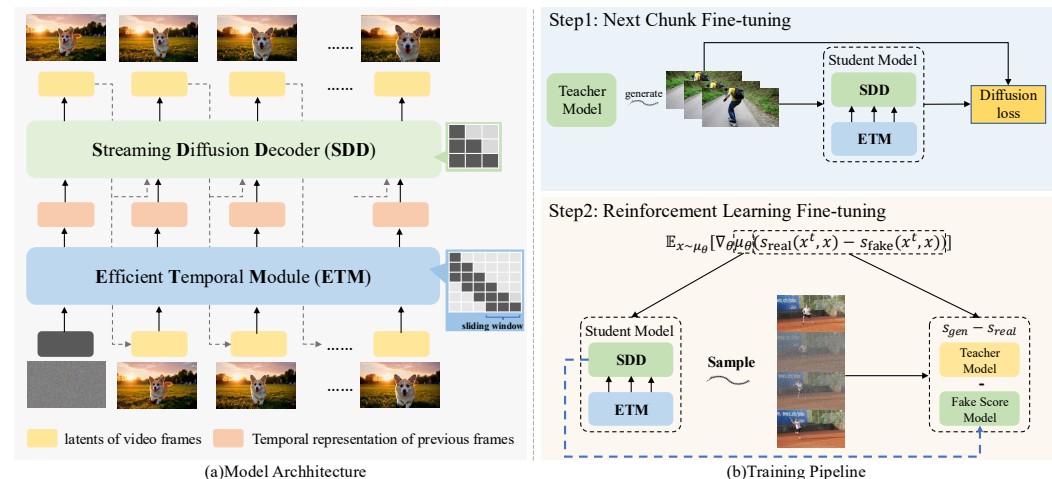

Figure 3: **Greedy Distill** comprises two main components: **E**fficient **T**emporal **M**odule (**ETM**) and **S**treaming **D**iffusion **D**ecoder (**SDD**) as shown (a).And training pipeline comprises two key stages: Next Block Fine-tuning and RL Fine-tuning.The score function $s$ is defined in Sec.2.2.2.

imperfect predictions at inference time. The rollout paradigm in RL effectively tackles this issue, as policy gradients are directly applied to the model's inner predictions throughout the entire generation process, thus reducing the reliance on ground-truth context during inference.

We apply Greedy Distill to the Wan2.1 video diffusion model, reducing the latency time to 0.24 s and achieving **a speed-up of** $\times$**2**. On the Wan2.1 1.3B model, **inference speed reaches 24 FPS**, enabling real-time, high-fidelity video synthesis for interactive applications. Experiments show that Greedy Distill attains few-step quality comparable to the multi-step teacher while offering stronger interactivity and faster generation. To our knowledge, this is the first distillation paradigm that allows substantial architectural differences between teacher and student models.

We provide a detailed review and discussion of related work in Appendix B.1.

## 2 METHODOLOGY

**Greedy Distill** introduces a new asymmetric distillation framework 3, which distills a pretrained bidirectional video diffusion model as teacher model into an efficient student model comprising two main components: **E**fficient **T**emporal **M**odule (**ETM**) and **S**treaming **D**iffusion **D**ecoder (**SDD**). Specifically, ETM employs an autoregressive transformer with sliding window attention to capture local and global features and produces a temporal representation, while SDD follows the diffusion paradigm to generate the next frame in a streaming manner conditioned on the first frame, the last frame and the temporal representation.

The framework overview and the training pipeline is shown in Figure. 3, comprises two key stages: Next Block Fine-tuning (Sec.2.2.1) and Reinforcement Learning Fine-tuning (Sec.2.2.2). Notably, both ETM and SDD are trained via Low Rank Adaptation (LoRA) (Hu et al., 2022) layers and initialized from the teacher. We also enable efficient inference using the sliding window attention mechanism and KV caching.

### 2.1 MODEL ARCHITECTURE

#### 2.1.1 EFFICIENT TEMPORAL MODULE(ETM)

We begin by compressing the video into a latent space using a 3D VAE. The VAE encoder processes each chunk of video frames independently, compressing them into shorter chunks of latents. The decoder then reconstructs the original video frames from each latent. Our causal diffusion transformer operates in this latent space, generating latents sequentially.

Unlike common AR models (Radford et al., 2019; Brown et al., 2020), ETM employ a chunk-wise sliding window attention mechanism inspired by prior work that combines autoregressive models with diffusion (Zhen et al., 2025). Within each chunk, we apply bidirectional attention among latents. To capture both local and global dependencies while controlling computational cost, we employ sliding window attention across chunks of latents. Formally, the attention mask $M$ of chunk-wise sliding window attention is typically defined as:

$$M_{i,j} = \begin{cases} 1, & \text{if } \left\lfloor \dfrac{i}{k} - w \right\rfloor \leq \left\lfloor \dfrac{j}{k} \right\rfloor \leq \left\lfloor \dfrac{i}{k} \right\rfloor, \\ 0, & \text{otherwise.} \end{cases} \tag{1}$$

where $i$ and $j$ index of latents of input frames, $k$ is the chunk size, $w$ is a fixed window size, and $\lfloor \cdot \rfloor$ denotes the floor function. ETM follows an autoregressive paradigm, compressing historical latents into a temporal representation:

$$c_f = \text{ETM}_\theta((x_0, x_1, ..., x_{f-1}), Mask = M) \tag{2}$$

where $c_f$ denotes a temporal representation of the index of $f$ of the latents of input frames, and $x_i$ denotes the index of $i$ of the latents. In this way, ETM can capture strong local and weak global cues to form a temporal representation while avoiding the cost of full global attention.

### 2.1.2 STREAMING DIFFUSION DECODER(SDD)

We then apply SDD to generate the next frame in a streaming manner. Concretely, **SDD** employs DiT with Flow Matching (Lipman et al., 2022; Liu et al., 2022), which assumes that the trajectory connecting a data sample $x$ and a noise sample $\epsilon$ in latent space follows a straight-line path:

$$x^t = (1 - t) \cdot x + t \cdot \epsilon \tag{3}$$

where $\epsilon \sim \mathcal{N}(0, I)$ and $t \in [0, 1]$. **SDD** learns a transformation from noise sample to samples drawn from the data distribution, formulated through an Ordinary Differential Equation (ODE):

$$\frac{dx^t}{dt} = \text{SDD}_\theta(x^t, \, t, \, c_f) \tag{4}$$

Here, $\text{SDD}_\theta$ represents a learnable velocity field parameterized by the model weights $\theta$. $t \in [0, 1]$ denotes the continuous time variable and $x^t$ refers to the data point at time $t$. $c_f$ is the temporal representation generated by ETM. Putting Eq. 2 and Eq. 4 all together, the forward process is:

$$G_\theta \triangleq \hat{x}_{1:F} = \{\Psi_{T:1}(\text{SDD}_\theta, \, t, \, c_f) | f = 1, 2, ..., F\} \tag{5}$$

where $G_\theta$ denotes video generator which uses $SDD_\theta$ and $c_f$ from ETM to autoregressively generate all latents of frames. $\hat{x}_f$ denotes the predicted latents at index $f$, $\Psi$ denotes the integrator (*i.e.*UniPC Solver (Zhao et al., 2023)) , that simulates the forward diffusion process from $T$-th step to 0-th step to get $x_0$.

### 2.1.3 INFERENCE PROCESS

During inference, both SDD and ETM in our model architecture utilize the KV cache strategy to make the inference more efficient. Specifically: ① When using ETM for inference, we use the key and value of the previous $w$ chunks to predict the next chunk. Therefore, the computational complexity of this part of our inference is $(L^2) \times w$ for one chunk, where L is the feature length in the latent space for one chunk. ② When using SDD for inference, we use the key and value of the previous chunk to predict the next chunk. Therefore, the computational complexity of this part of our inference is $(L^2) \times T$ for one chunk, where the number of sampling steps is T.

Therefore, the overall time with linear time complexity can be expressed as the sum of the two components is $(w + T) \times (L^2) \times F$, where $F$ is the number of chunks of video, the feature length($L$) is fixable for given teacher model. Finally, the inference cost of Greedy Distill is a linear time complexity that is only dependent on $F$. Complete description refers to the Algorithm 1.

---

**Algorithm 1** Inference Process with KV Caching

---

**Require:** Denoising timesteps $\{t_0 = 0, t_1, \ldots, t_Q\}$, video length $F$, few-step autoregressive video generator $G_\theta$, sliding window size $w$
1: **Initialize** KV cache $C_{\text{ETM}} \leftarrow \emptyset$, $C_{\text{SDD}} \leftarrow \emptyset$
2: **Initialize 0-th frame:** $x_0 \sim \mathcal{N}(0, I)$
3: **for** $f = 1$ to $F$ **do**
4:     **Generate temporal representation:** $c_f = ETM_\theta(x_{f-w}, \ldots, x_{f-1})$ using cache $C_{ETM}$
5:     Append new KV pairs to cache $C_{\text{ETM}}$
6:     **if** $f > w$ **then**
7:         Remove oldest KV pairs
8:     **end if**
9:     **Initialize current frame:** $x_f^T \sim \mathcal{N}(0, I)$
10:     **for** $t = T$ to 1 **do**
11:         **Generate current frame:** $x_f^t = \Psi_{T:t}(\text{SDD}_\theta, \ t, \ c_f)$ using cache $C_{SDD}$
12:         **if** $t = T$ **then**
13:             Append new KV pairs to cache $C_{\text{SDD}}$
14:         **end if**
15:         **if** $t = 1$ **then**
16:             Clear KV cache $C_{SDD}$
17:         **end if**
18:     **end for**
19:     $x_f = x_f^0$
20: **end for**
21: **Return** $\{x_f\}_{f=1}^F$

---

## 2.2 TRAINING PIPELINE

### 2.2.1 NEXT CHUNK FINE-TUNING

We follow the common *next-token fine-tuning* paradigm in autoregression, but extend it here to *next-chunk fine-tuning*. SDD generates the next frame in a streaming manner, conditioned on the first frame, the last frame and the temporal representation from ETM. The loss function is defined as:

$$\mathcal{L}_{\text{nc}} = \mathbb{E}_{t \sim \text{Uniform}([0,1]), x \sim P} \frac{1}{F} \sum_{f=1}^{F} \left\| (\epsilon - x_f) - \text{SDD}_\theta(x_f^t, \ t, \ \text{ETM}_\theta(x_0, x_1, \ldots, x_{f-1})) \right\|^2 \tag{6}$$

### 2.2.2 ADDRESSING ERROR ACCUMULATION WITH REINFORCEMENT LEARNING

In experiments (Tab. 3), we observe that after *next-chunk fine-tuning*, the model already demonstrates certain generative capabilities, but it still suffers from **error accumulation**. In addition, qualitative results reveal that some frames remain insufficiently sharp after next-chunk fine-tuning.

Generally, error accumulation problem is more broadly known as exposure bias, where a model is trained exclusively on ground-truth context but must rely on its own imperfect predictions at inference time, resulting in a distributional mismatch that compounds errors as generation progresses. To mitigate this issue, we make **the first attempt to incorporate RL fine-tuning** into the distillation process. The key advantage of RL lies in its actor–critic architecture and rollout paradigm, which directly optimizes the model's own predictions across the entire generation process. By doing so, RL effectively alleviates error accumulation by reducing the model's reliance on ground-truth context at inference time, thereby addressing the exposure bias problem.

Here, we use **deterministic policy gradient** (Lillicrap et al., 2015; Fujimoto et al., 2018), where the policy gradient is defined as:

$$\nabla_\theta \mathcal{J} = -\mathbb{E}_{s \sim \rho^\mu} \left[ \nabla_\theta \mu_\theta(s) \, \nabla_a Q^\mu(s, a)|_{a = \mu_\theta(s)} \right] \tag{7}$$

where $\mu_\theta$ is a parameterized actor function which specifies the current policy by deterministically mapping states to a specific action. The critic $Q(s, a)$ is learned using the Bellman equation (Bellman, 1954) as in Q-learning (Watkins & Dayan, 1992).

Specifically, in our approach, we define the expected reward as the KL divergence $KL(P_{gen}(x)|P_{real}(x))$. It can be proven that (refer to the Appendix C):

$$\nabla_\theta \mathcal{J} \propto -\mathbb{E}_{x_t \sim \rho^\mu} \left[ \nabla_\theta \mu_\theta(x_t) \cdot \mathbb{E}_{p_{fake}} \nabla_{x_{t-1}} \log \frac{p_{\text{fake}}(x_{t-1}|x_t)}{p_{\text{real}}(x_{t-1}|x_t)} \right] \tag{8}$$

Following the derivations in DMD (Yin et al., 2024b), we rewrite $\nabla_{x_{t-1}} \log \frac{p_{\text{fake}}(x_{t-1}|x_t)}{p_{\text{real}}(x_{t-1}|x_t)}$ as $s_{gen}(x_{t-1}, t-1) - s_{real}(x_{t-1}, t-1))$, where $s$ denotes the score function. In practice, we approximate the score function by the velocity field $\psi_\theta$ as DMD. Combining Equations 8 and 5, the overall loss of reinforcement learning fine-tuning is:

$$\nabla_\theta \mathcal{J} \propto -\mathbb{E}_{x \sim \mu_\theta} \left[ \nabla_\theta \mu_\theta(x_t^f) \left( s_{gen}(x_{t-1}, t-1) - s_{real}(x_{t-1}, t-1) \right) \right] \tag{9}$$

where $x_{t-1} = \mu_\theta(x_t^f) = \Psi_{T:t}(\text{SDD}_\theta(x_f^t, t, \text{ETM}_\theta(x_0, x_1, \ldots, x_{f-1})))$, $s_{gen}, s_{real}$ are score function, while $s_{gen}$ is provided by the $\text{SDD}_\theta$ and $s_{real}$ corresponds to the teacher model. The RL fine-tuning loss $\mathcal{L}$ is formulated as:

$$\mathcal{L}_{RL} = \mathbb{E}_{x \sim \mu_\theta, f \in [1,2,\ldots F], t \in \mathcal{T}} \text{MSE} \left[ \mu_\theta(x_t^f) - \text{sg}[\mu_\theta(x_t^f) - (s_{real}(x_{t-1}^f, t-1) - s_{gen}(x_{t-1}^f, t-1))] \right] \tag{10}$$

where $\mathcal{T}$ denotes the set of steps in our distillation objective which is set as $[1000, 750, 500, 250]$, MSE denotes Mean Squared Error, and $sg$ denotes the stop-gradient operation.

The key difference between our method and DMD is that DMD's score function relies on training another network, while ours is derived from $\text{SDD}_\theta$. Additionally, DMD's critic model update incurs high computational costs, whereas our method updates both the policy and critic models simultaneously without additional training, significantly reducing computational costs. The overall training of our framework is done in two steps, the model is optimized with $\mathcal{L}_{nc}$ at the first stage while the $\mathcal{L}_{RL}$ is used at the second stage for further fine-tuning.

## 3 EXPERIMENT

### 3.1 SETUP

**Implementation.** Our teacher model is based on the bidirectional DiT architecture of Wan2.1, which processes video data in the latent space. A 3D VAE is used to compress video frames into latents. The student model adopts our Greedy Distill architecture, consisting of ETM and SDD. Both components are initialized from the Wan2.1 and fine-tuned via Low Rank Adaptation (LoRA) (Hu et al., 2022). The key distinction is that ETM is a causal attention mechanism combined with a sliding-window mechanism to capture the global temporal information across frames, which substantially reduces computational cost. The temporal representation produced by ETM is able to attend not only to the current window but also to information from preceding windows, thereby balancing efficiency with temporal context modeling.

**Training and Inference.** Our training process consists of two stages: Next Block Fine-tuning (Sec.2.2.1) and Reinforcement Learning Fine-tuning (Sec.2.2.2). The prompts for training are taken from OpenVidHD (Nan et al., 2024), while the video data are sampled from the teacher model, i.e., WAN 2.1 (Wan et al., 2025).

The model is trained on 64 NVIDIA H100 GPUs using the AdamW optimizer. The input video resolution is set to $832 \times 480$, and each video contains between 81 and 301 frames. In the Next Block Fine-tuning stage, the student model is trained for 10 epochs with a batch size of 1024 and a learning rate of $1 \times e^{-5}$. In the Reinforcement Learning Fine-tuning stege, the student model is trained for 3 epochs with a batch size of 256 and a learning rate of $2 \times e^{-6}$.

**Evaluation metrics.** We conduct a comprehensive evaluation of our method using VBench (Huang et al., 2024) as the primary metric. We rewrite the test prompts using Qwen/Qwen2.5-7B-Instruct (Team, 2024), a practice that has already been widely adopted in prior work(*i.e.*, Self Forcing (Huang et al., 2025) and OpenVid (Nan et al., 2024)). In addition, we perform human evaluation to assess the perceptual quality of the generated videos. We compare our approach against state-of-the-art distillation methods from multiple perspectives, and the aggregated results demonstrate that our method consistently outperforms competing approaches.

Table 1: We compare Greedy Distill with representative open-source video generation models of similar parameter sizes and resolutions.

| Model | #Params | Resolution | Throughput (FPS) ↑ | Latency (s) ↓ | Evaluation scores ↑ | | |
|---|---|---|---|---|---|---|---|
| | | | | | Total Score | Quality Score | Semantic Score |
| *Diffusion models* | | | | | | | |
| LTX-Video (HaCohen et al., 2024) | 1.9B | 768×512 | 8.98 | 13.5 | 80.00 | 82.30 | 70.79 |
| Wan2.1 (Wan et al., 2025) | 1.3B | 832×480 | 0.78 | 103 | 84.26 | 85.30 | 80.09 |
| *Chunk-wise autoregressive models* | | | | | | | |
| SkyReels-V2 (Chen et al., 2025b) | 1.3B | 960×540 | 0.49 | 112 | 82.67 | 84.70 | 74.53 |
| CausVid (Yin et al., 2025)* | 1.3B | 832×480 | 17.0 | 0.69 | 81.20 | 84.05 | 69.80 |
| Self Forcing(chunk-wise) (Huang et al., 2025) | 1.3B | 832×480 | 17.0 | 0.69 | 84.31 | 85.07 | 81.28 |
| *Frame-wise Autoregressive models* | | | | | | | |
| NOVA (Deng et al., 2024) | 0.6B | 768×480 | 0.88 | 4.1 | 80.12 | 80.39 | 79.05 |
| Pyramid Flow (Jin et al., 2024) | 2B | 640×384 | 6.7 | 2.5 | 81.72 | 84.74 | 69.62 |
| Self Forcing(frame-wise) (Huang et al., 2025) | 1.3B | 832×480 | 8.9 | 0.45 | 84.26 | 85.25 | 80.30 |
| Greedy Distill (Ours) | 1.3B | 832×480 | **24.0** | **0.24** | **84.60** | **85.37** | **81.52** |

## 3.2 MAIN RESULTS

**Quantitative Comparison.** To ensure fairness, all experimental results are obtained from the same model scale, *i.e.*, 1.3B-2.0B. Table 1 presents a comprehensive comparison between Greedy Distill and existing state-of-the-art methods. Obviously, our method achieves the **highest VBench score**, while meets the **real-time** requirements, *i.e.*, 24 FPS in Throughput and a Latency Time of only 0.24 seconds. Moreover, Greedy Distill maintains generation quality comparable to that of the teacher model (**85.60** vs. 85.26 of Wan2.1-1.3B).

**Qualitative Comparison.** We compare the videos generated by our method with those from CausVid (Yin et al., 2025) and Self Forcing (Huang et al., 2025), as in Figure. 10. The results indicate that CausVid and Self Forcing suffers from inconsistencies with physical dynamics and CausVid in particular is prone to error accumulation, whereas Greedy Distill maintains high-quality video generation while ensuring fast inference speed, effectively avoiding the error accumulation problem and adhering more closely to physical principles.

Specifically, as shown in Figure. 10 (a) and Figure. 10 (b), CausVid and Self Forcing produce object details that deviate from their natural properties (e.g., unrealistic cycling postures and incorrect numbers of cat tails). In Figure. 10 (c) andFigure. 10 (d), the generated videos from CausVid and Self Forcing violate physical dynamics (e.g., a ball remaining static in midair or an airplane following an implausible curved trajectory during landing). In contrast, our method does not suffer from these issues, producing results that are both physically consistent and visually coherent.

**User Study.** To further evaluate the perceptual quality of videos generated by our method, we conducted a human assessment study. For each model, we selected 40 video samples of varying durations to reflect challenges across different video lengths: 15 short clips (0–5 seconds), 15 medium-length clips (5–10 seconds), and 10 long clips (10–18 seconds). The evaluation involved 60 participants (50% aged 18–30, 30% aged 30–40, and 20% aged 40–50; 41.7% male, 58.3% female). Each participant was presented with a text prompt alongside videos generated by different models, with all videos displayed in random order to minimize ordering bias. Following prior work (Kong et al., 2024), participants were asked to select the video they perceived as better in terms of text alignment, motion quality, and visual quality. As shown in Fig. 5, Our videos are preferred over others, especially in text alignment and motion quality.

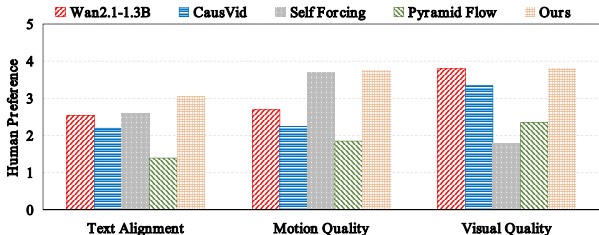

Figure 5: **User preference study.** Greedy Distill outperforms all baselines in human preference.

## 3.3 ABLATION STUDY

**Long Video Generation.** To demonstrate that our method's ability in mitigating the error accumulation problem, we conduct long-video generation experiments (*i.e.*, 18s). From the qualitative results(*i.e.*, Figure. 7) on long video generation, we also observe that both CausVid and Self-Forcing

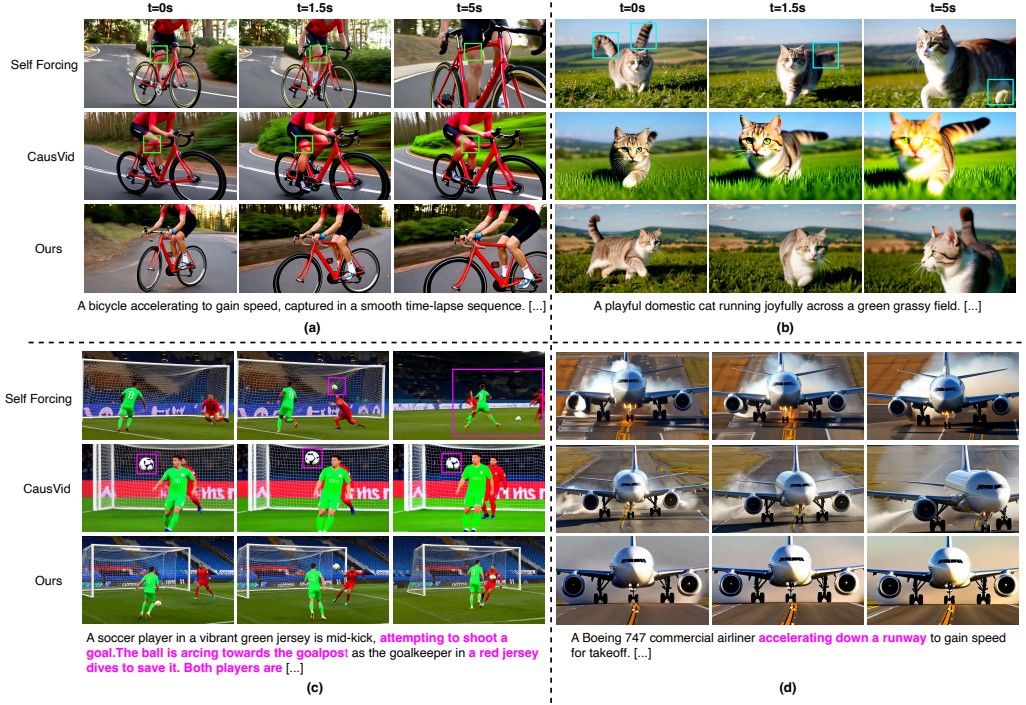

Figure 4: **Qualitative comparisons** between Greedy Distill and Self Forcing(chunk-wise), CausVid. All models are distilled from the same teacher model of Wan2.1.*Videos are available in the supplementary materials.*

Table 2: Evaluation of long video generation.

| Method | Throughput (FPS) | Temporal Coherence | Frame Quality | Semantic Alignment |
|---|---|---|---|---|
| Streaming T2V (Henschel et al., 2025) | 0.5 | 88.9 | 45.3 | 27.1 |
| CausVid (Yin et al., 2025) | 17.0 | 88.5 | 60.1 | 25.8 |
| Self Forcing (Huang et al., 2025) | 17.0 | 90.3 | 61.3 | 26.9 |
| **Greedy Distill (Ours)** | 24.0 | **94.2** | **61.7** | **27.7** |

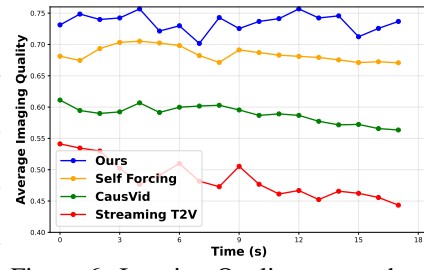

Figure 6: Imaging Quality scores show that **Greedy Distill** maintains stable and superior image quality in long video.

suffer from error accumulation and produce videos with low dynamics or static content in the later segments. For example, after the 9-second mark, the main objects in videos generated by CausVid and Self-Forcing either remain motionless or exhibit only minor movements. In addition, these methods suffer from over-exposure artifacts caused by accumulated errors. Our method shows a clear advantage(*i.e.*Table. 2) on long-video evaluation with VBench, particularly in the **Temporal Coherence** metric, where it achieves an improvement of **4.3%**. This demonstrates that our approach not only maintains frame-level fidelity but also better preserves consistency across extended time horizons, effectively addressing the error accumulation and stagnation issues observed in prior methods, as shown in Fig. 6,

**Differences between Greedy Distill and previous approaches.** We compare Greedy Distill with prior approaches in two aspects. ① **Teacher-Student Model Architecture:** As in Figure 8, previous methods use nearly identical teacher and student models, with only a causal attention mechanism, incurring significant computational overhead of $T \times w \times F \times L^2$, where $L$ is the sequence length. In contrast, Greedy Distill introduces a novel student architecture and reduce complexity to $(T + w) \times F \times L^2$. The ETM effectively captures global features for temporal representations while avoiding the high cost of global attention. ② Previous methods generally adopt a **chunk-wise inference** strategy, outputting results every 3 chunks and resetting the context every 7 chunks. This design, however, leads to severe error accumulation and temporal artifacts. While some approaches attempt to mitigate these issues through diffusion forcing (*i.e.*CausVid, SkyReels-V2) or self forcing (*i.e.*Self Forcing), the problems remain significant (shown in Figure. 7). In contrast, our method leverages a

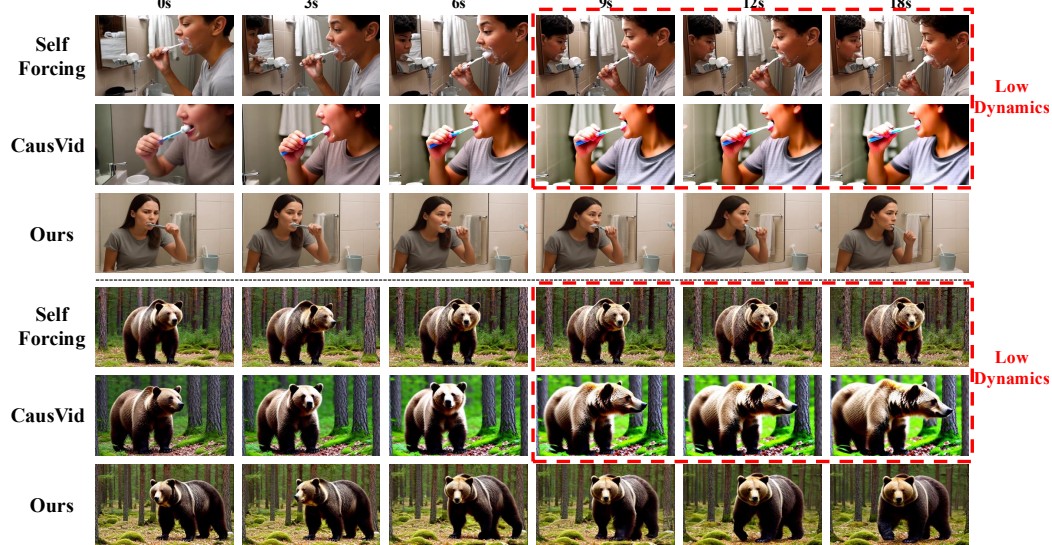

Figure 7: Qualitative results on long video generation show that our method avoids error accumulation and low-dynamics issues. *Videos are available in the supplementary materials.*

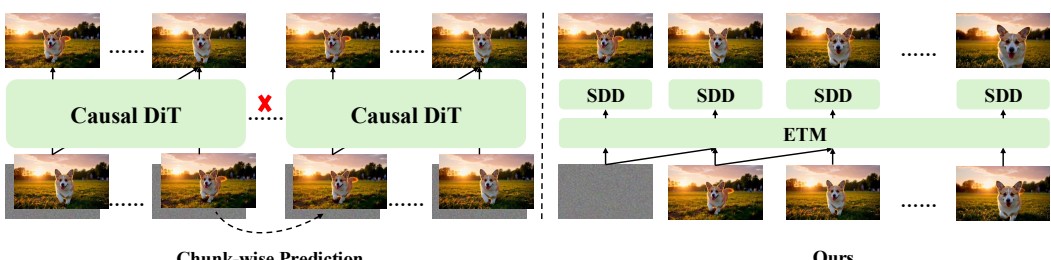

Figure 8: Differences between Greedy Distill and prior approaches.

ETM module, which allows the decoder to access global information, effectively avoiding the error accumulation problem and adhering more closely to physical principles.

**Necessity of ETM and RL Fine-tuning.** We investigate the need for ETM in the student model of Greedy Distill and Reinforcement Learning Fine-tuning, as shown in Table 3. Removing ETM and performing direct distillation (similar to CausVid and Self-Forcing with a chunk size of 1) reveals the limitations of using only a diffusion decoder in the student model. Introducing ETM with sliding-window attention improves Quality Score (+3.3%). Finally, RL Fine-tuning yields the best results, improving Quality Score by 5.73% and mitigating error accumulation. Meanwhile, the training loss curve of RL Fine-tuning, which corresponds to the action-value, is illustrated in Fig. 9.

Table 3: **Ablation studies.** We compares different components in our distillation framework..The last row is our final configuration

| RL | ETM | Total Score | Quality Score | Semantic Score |
|---|---|---|---|---|
| ✗ | ✗ | 81.01 | 80.93 | 81.32 |
| ✗ | ✓ | 83.2 | 83.63 | 81.49 |
| ✓ | ✓ | **84.60** | **85.37** | **81.52** |

## 4 CONCLUSION

In this paper, we present GREEDY DISTILL, a novel distillation training paradigm for autoregressive video diffusion models. It employs the Streaming Diffusion Decoder (SDD) to generate intermediate frames using only the 0-th and the last frames, avoiding redundant computation, and the Efficient Temporal Module (ETM) to capture global temporal dependencies. GREEDY DISTILL reduces the computational complexity from $O(n^2)$ to linear. Moreover, for the first time, we applies reinforcement learning fine-tuning to mitigate error accumulation in streaming generation. Our approach achieves strong improvements in both real-time and long-duration video generation.

## REPRODUCIBILITY STATEMENT

This statement presents a comprehensive report detailing the reproduction process for our Greedy Distill, a distillation training paradigm for autoregressive video diffusion models. The implementation builds upon Diffusers's code base and integrates components from additional open-source libraries, to which we extend our gratitude.

### KEY IMPLEMENTATION DETAILS

- **Code Base**: In terms of implementation, we build upon the **Diffusers** project as our code base, using /src/diffusers/models/transformers/transformer_wan.py as the foundation for the DiT architecture. For Low-Rank Adaptation (LoRA), we leverage the open-source **PEFT** library as an additional component.

- **SDD Implementation**: The SDD is initialized from Wan2.1 and fine-tuned using Low-Rank Adaptation (LoRA). Furthermore, the teacher model's global attention is replaced with **causal attention**, enabling streaming inference and leveraging KV cache to improve inference efficiency.

- **ETM Implementation**: The ETM is initialized from Wan2.1 and fine-tuned using Low-Rank Adaptation (LoRA). In this process, the teacher model's global attention is replaced with **causal sliding-window attention** (window size set to 3), enabling streaming inference and leveraging KV cache to further improve efficiency.

- **RL Fine-tuning Implementation**: In the RL Fine-tuning stage, we randomly sample $f \in [1, 2, ...F], t \in \mathcal{T}$, compute the loss according to Eq. 10, and adopt a smaller learning rate. Notably, in distributed training, to improve GPU utilization, we enforce the same $f$ and $t$ within each minibatch. The student model is trained for 3 epochs with a batch size of 256 and a learning rate of $2 \times e^{-6}$. The training loss curve is illustrated in Fig. 9.

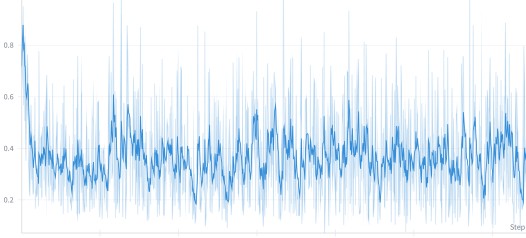

Figure 9: The training loss curve in RL Fine-tuning.

### RESULTS

Using the aforementioned process, we successfully distill the Wan2.1 model into a student model that meets real-time requirements, achieving 24 FPS throughput and a VBench score of 84.60.

### CONCLUSION

This reproduction report documents the end-to-end procedure for replicating Greedy Distill on Wan2.1, covering codebase choices (Diffusers/PEFT), module initialization (SDD/ETM via LoRA), and training protocols (Next-Block and RL fine-tuning). We provide exact hyperparameters, data sources, and inference settings—including causal sliding-window attention with KV cache—to enable faithful re-creation. Our reimplementation attains 24 FPS throughput and a VBench score of 84.60, validating the method's reproducibility. We release scripts and configuration files to streamline replication and adaptation to other backbones or hardware budgets. These artifacts offer a robust foundation for advancing efficient, real-time, long-duration video generation research.

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

## A   THE USE OF LARGE LANGUAGE MODELS(LLMS)

In preparing this paper, large language models (LLMs) were used solely for language refinement, such as improving grammar, clarity, and fluency. All research questions, conceptual and theoretical frameworks, methodology, data analysis, and conclusions were developed and carried out independently by the author. The LLMs did not generate or influence any core ideas, interpretations, or findings. Their role was limited to enhancing readability while preserving the originality and integrity of the scholarly work.

## B   RELATED WORK

### B.1   AUTOREGRESSIVE VIDEO GENERATION

Autoregressive video generation aims to synthesize videos frame by frame along the temporal dimension, thereby achieving lower latency and improved temporal coherence. Inspired by the remarkable success of large language models (LLMs) (Bai et al., 2023) in natural language processing, early studies (Yan et al., 2021; Ge et al., 2022; Wu et al., 2022; Kondratyuk et al., 2023; Wang et al., 2024; Wu et al., 2024) encoded videos into discrete tokens and employed autoregressive Transformers to generate video tokens sequentially. More recently, diffusion models (Ho et al., 2020; Song et al., 2020; Lipman et al., 2022) have achieved significant advances in video generation and have been widely adopted in this domain. Some works (Alonso et al., 2024; Jin et al., 2024; Valevski et al., 2024; Zhang et al., 2024; Chen et al., 2024; Kim et al., 2024; Ruhe et al., 2024) train diffusion models to denoise new frames conditioned on the given context frames, thereby enabling autoregressive generation. Even more recently, a line of research has explored leveraging pre-trained text-to-image (Kodaira et al., 2023; Liang et al., 2024; Valevski et al., 2024; Weng et al., 2024) or text-to-video (Gao et al., 2024; Kim et al., 2024; Xing et al., 2024; Xie et al., 2025; Henschel et al., 2025) models and adapting them to perform autoregressive next-frame generation conditioned on context frames. Our approach is closely related to this research direction. The key difference is that we propose an innovative adaptation method via diffusion distillation. This method not only significantly improves efficiency but also makes autoregressive methods competitive with bidirectional diffusion in video generation.

### B.2   DIFFUSION MODEL DISTILLATION

Diffusion distillation aims to distill knowledge from a pre-trained teacher diffusion model to a student diffusion model, enabling the student to generate samples in fewer steps and thereby reducing inference costs. According to the distillation mechanism, previous studies can be roughly divided into two categories: trajectory-preserving distillation and Distribution-matching distillation. Trajectory-preserving distillation aims to predict the ordinary differential equation (ODE) trajectory of the teacher model with fewer steps. Luhman & Luhman (2021); Zheng et al. (2023) trained the student model on noise–image pairs precomputed by the teacher model using an ODE solver. Progressive distillation (Meng et al., 2023; Salimans & Ho, 2022) trains a series of student models, iteratively halving the number of sampling steps at each stage to reduce the total number of steps required. instaflow (Liu et al., 2022; 2023) uses a reflow-based distillation approach to align noise and image mappings, enabling accurate one-step generation. Consistency Distillation (Luo et al., 2023a;b; Song & Dhariwal, 2023; Song et al., 2023; Gu et al., 2023; Berthelot et al., 2023; Kim et al., 2023; Zheng et al., 2024a; Lu & Song, 2024; Liu et al., 2025) trains student models to produce outputs aligned with the teacher at all timesteps along the ODE trajectory, thereby achieving self-consistency. Unlike trajectory-preserving distillation, which imposes constraints via ODE paths, distribution-matching distillation supervises at the distributional level, aligning the output distributions of the student and teacher models. Some approaches (Xiao et al., 2021; Kang et al., 2024; Luo et al., 2024b; Sauer et al., 2024a;b; Xu et al., 2024; Chen et al., 2025a; Lin et al., 2025) reduce the distribution discrepancy through adversarial training, while others (Luo et al., 2023c; 2024a; Yin et al., 2024a;b;c; Zhou et al., 2024) achieve this via score-based distillation.

In contrast to aforementioned approaches that primarily focus on distillation through reducing the number of generation steps, we propose a novel student distillation architecture. Our approach not only reduces the number of generation steps but also decreases the computational complexity along

the frame dimension from quadratic to linear. In particular, compared with the asymmetric strategy adopted by CausVid (Yin et al., 2025), which distills a bidirectional teacher into a unidirectional student, our method introduces a more substantial structural innovation by integrating autoregressive (AR) and diffusion paradigms. This fusion design achieves higher inference efficiency while preserving generation quality.

## C  DERIVATION FOR REINFORCEMENT LEARNING FINE-TUNING

We frame the training objective within a reinforcement learning paradigm. We consider the denoising process as a Markov Decision Process (MDP). We define $\bar{t}$ to be the timestep of MDP, and the relation between MDP timestep $\bar{t}$ and denoising timestep $t$ is $\bar{t} = T - t$. Let $s_{\bar{t}} \in \mathcal{S}$ to be the state at MDP timestep $\bar{t}$, where $\mathcal{S}$ is the state space. We define $s_{\bar{t}} = x_{T-\bar{t}} = x_t$, which is the noisy image $x_t$ at timestep $t$. So when $\bar{t} = 0$, $t = T$, and we have $s_{\bar{t}=0} = x_T$, which is the initial noise. When $\bar{t} = T$, $t = 0$, and we have $s_{\bar{t}=T} = x_0$, which is the fully denoised image. $a_{\bar{t}} \in \mathcal{A}$ is the action at timestep $\bar{t}$, where $\mathcal{A}$ is the action space. We define $a_{\bar{t}}$ to be the noise $\epsilon_t$ added to noise image $x_t$. Thus we have $s_{\bar{t}+1} = x_{t-1}$, because of the claim $\bar{t} = T - t$.

The expected value of cumulative return along the entire trajectory $\tau = (x_T, x_{T-1}, \ldots, x_0)$ can be defined as:

$$
\begin{aligned}
E_{\tau \sim \rho}(R(\tau)) &= D_{KL}\left(p_{\text{fake}}(\tau) \parallel p_{\text{real}}(\tau)\right) \\
&= E_{x \sim p_{\text{fake}}} \log \frac{p_{\text{fake}}(x_{t=0})}{p_{\text{real}}(x_{t=0})} \\
&= E_{x \sim p_{\text{fake}}} \sum_{t=1}^{T} \log \frac{p_{\text{fake}}(x_{t-1}|x_t)}{p_{\text{real}}(x_{t-1}|x_t)} \\
&= \sum_{t=1}^{T} E_{x \sim p_{\text{fake}}} R(x_{t-1}, x_t)
\end{aligned}
\tag{11}
$$

Thus, we define the $r(s_{\bar{t}}, a_{\bar{t}})$ as the log-ratio of transition probabilities under the fake and real distributions:

$$
r(s_{\bar{t}}, a_{\bar{t}}) = R(x_{t-1}, x_t) = \log \frac{p_{\text{fake}}(x_{t-1}|x_t)}{p_{\text{real}}(x_{t-1}|x_t)}
\tag{12}
$$

This represents the log-ratio of the probabilities of the entire trajectory under the fake and real distributions.

The action-value function (Q-function) for taking action $x_{t-1}$ in state $x_t$ and following the policy thereafter is defined as:

$$
\begin{aligned}
Q(s_{\bar{t}}, a_{\bar{t}}) &= \mathbb{E}_{p_{\text{fake}}} \sum_{k=\bar{t}}^{T} r(s_{\bar{t}}, a_{\bar{t}}) \\
&= \mathbb{E}_{p_{\text{fake}}} \sum_{k=1}^{t} \log \frac{p_{\text{fake}}(x_{k-1}|x_k)}{p_{\text{real}}(x_{k-1}|x_k)}
\end{aligned}
\tag{13}
$$

The defined Q-function satisfies the Bellman equation for the expected return:

$$
\begin{aligned}
Q(s_{\bar{t}}, a_{\bar{t}}) &= \mathbb{E}_{p_{\text{fake}}} \sum_{k=1}^{t} \log \frac{p_{\text{fake}}(x_{k-1}|x_k)}{p_{\text{real}}(x_{k-1}|x_k)} \\
&= \mathbb{E}_{p_{\text{fake}}} \left[ \log \frac{p_{\text{fake}}(x_{t-1}|x_t)}{p_{\text{real}}(x_{t-1}|x_t)} + \mathbb{E}_{p_{\text{fake}}} \sum_{k=1}^{t-1} \log \frac{p_{\text{fake}}(x_{k-1}|x_k)}{p_{\text{real}}(x_{k-1}|x_k)} \right] \\
&= \mathbb{E}_{p_{\text{fake}}} \left[ r(s_{\bar{t}}, a_{\bar{t}}) + Q(s_{\bar{t}+1}, a_{\bar{t}+1}) \right]
\end{aligned}
\tag{14}
$$

Instead of parameterizing policy directly as $\pi(a_{\bar{t}}|s_{\bar{t}})$, we parameterize the policy with distillation model $x = G_\theta(\epsilon), \epsilon \sim \mathcal{N}(0; \mathbf{I})$. The training objective can now be expressed as:

$$
\theta = \arg \max_{\theta} \mathcal{J}(\theta)
\tag{15}
$$

where
$$\mathcal{J}(\theta) = \mathbb{E}_{\tau \sim p_{\text{fake}}} \left[ R(\tau) \right] = D_{KL} \left( p_{\text{fake}}(\tau) \parallel p_{\text{real}}(\tau) \right) \tag{16}$$

Follow DDPG Lillicrap et al. (2015), the gradient of this objective with respect to the generator parameters $\theta$ is:

$$
\begin{aligned}
\nabla_\theta \mathcal{J} &= -\mathbb{E}_{s \sim \rho^\mu} \left[ \nabla_\theta \mu_\theta(s) \, \nabla_a Q^\mu(s, a) \big|_{a = \mu_\theta(s)} \right] \\
&= -\mathbb{E}_{x_t \sim \rho^\mu} \left[ \nabla_\theta \mu_\theta(x_t) \cdot \nabla_{a_{\bar{t}}} x_{t-1} \cdot \nabla_{x_{t-1}} Q(x_t, a_{\bar{t}}) \right]
\end{aligned}
\tag{17}
$$

Since $a_{\bar{t}}$ is the noise and $x_{t-1}$ is the next denoised image, so $\nabla_{a_{\bar{t}}} x_{t-1}$ is a constant, so we have:

$$\nabla_\theta \mathcal{J} = C \cdot -\mathbb{E}_{x_t \sim \rho^\mu} \left[ \nabla_\theta \mu_\theta(x_t) \cdot \nabla_{x_{t-1}} Q(x_t, a_{\bar{t}}) \right]$$

$$= C \cdot -\mathbb{E}_{x_t \sim \rho^\mu} \left[ \nabla_\theta \mu_\theta(x_t) \cdot \nabla_{x_{t-1}} \mathbb{E}_{p_{fake}} \sum_{k=1}^{t} \log \frac{p_{\text{fake}}(x_{k-1}|x_k)}{p_{\text{real}}(x_{k-1}|x_k)} \right] \tag{18}$$

$$= C \cdot -\mathbb{E}_{x_t \sim \rho^\mu} \left[ \nabla_\theta \mu_\theta(x_t) \cdot \mathbb{E}_{p_{fake}} \nabla_{x_{t-1}} \log \frac{p_{\text{fake}}(x_{t-1}|x_t)}{p_{\text{real}}(x_{t-1}|x_t)} \right]$$

## D  MORE DETAILS OF THE USER STUDY

To comprehensively evaluate the quality of generated videos, we conduct a user study based on a 5-point Likert scale (1 = Strongly Disagree, 2 = Disagree, 3 = Neutral, 4 = Agree, 5 = Strongly Agree). Participants were presented with paired videos generated by different models and were asked to score each video independently across three major dimensions: **Text Alignment, Motion Quality, and Visual Quality**. Each dimension contains multiple carefully designed questions targeting distinct aspects of video generation. The full questionnaire is provided below.

① **Text Alignment**

This dimension assesses how well the generated video matches the semantic intent of the input text prompt. Participants evaluated the following aspects:

**Object Class Accuracy**

The objects in the video correctly match the categories described in the prompt.

**Multiple Objects Handling**

The video correctly represents and maintains multiple objects without confusion or merging.

**Color Accuracy**

The colors in the video are accurate, stable, and consistent with the expected appearance.

**Spatial Relationship**

The spatial relationships between objects (e.g., relative positions, sizes) are logical and consistent.

**Scene Coherence**

The video presents a coherent and believable scene that aligns with the prompt.

**Appearance Style Consistency**

The appearance style (e.g., realistic, cartoon, cinematic) matches the intended style and remains stable.

② **Motion Quality**

This dimension focuses on the realism, stability, and smoothness of motion across the video sequence:

**Temporal Flickering**

The video shows minimal flickering or frame-to-frame jitter.

**Motion Smoothness**

Movements in the video appear smooth and natural without abrupt jumps.

**Dynamic Degree**

The video presents an appropriate level of dynamics that matches the scene and prompt.

**Temporal Style Consistency**

The stylistic elements of the video remain consistent across time.

**Overall Consistency**

The video maintains overall coherence and consistency across all frames.

③ **Visual Quality**

This dimension evaluates perceptual clarity, aesthetics, and frame-level consistency:

**Subject Consistency**

The main subject remains visually consistent throughout the video.

**Background Consistency**

The background stays stable and does not exhibit unexpected changes across frames.

**Aesthetic Quality**

The overall artistic and aesthetic quality of the video is appealing.

**Imaging Quality**

The video appears clear, sharp, and free from noticeable visual artifacts.

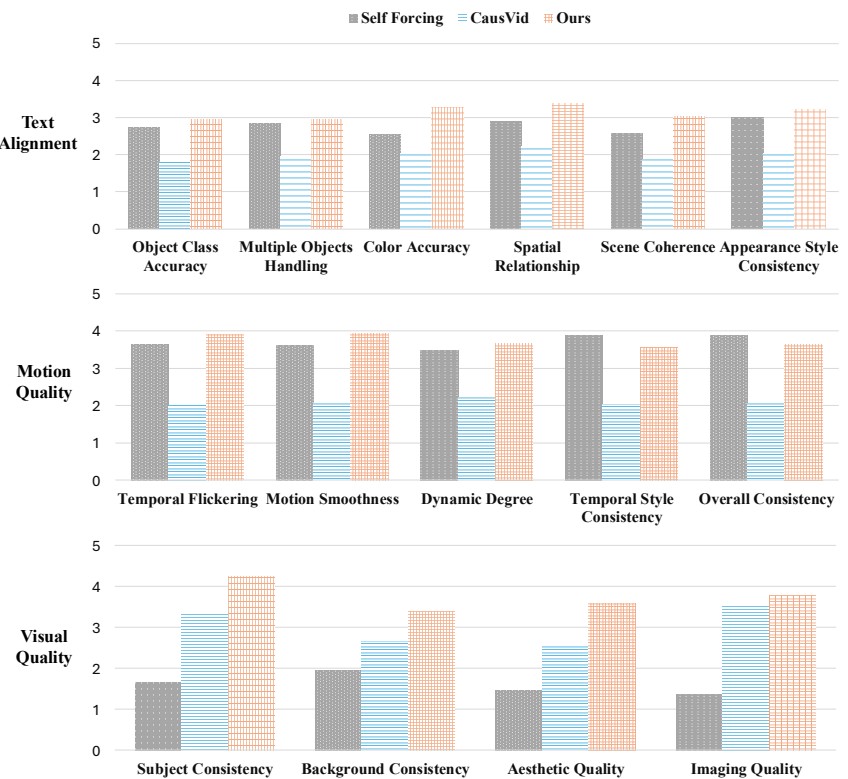

Figure 10: Detail scores of each dimension in the user study.

