# OpenReview forum: "Greedy Distill: Efficient Video Generative Modeling with Linear Time Complexity"
_ICLR.cc/2026/Conference — Submitted to ICLR 2026_

### Official Review · Reviewer_pNtk · 2025-10-29

**Soundness:** 3
**Presentation:** 2
**Contribution:** 2
**Rating:** 4
**Confidence:** 4

**Summary:**

The paper proposes Greedy Distill, an asymmetric distillation that turns a bidirectional DiT video diffusion teacher into a fast student composed of an Efficient Temporal Module (chunked AR Transformer with sliding-window attention) and a Streaming Diffusion Decoder with KV caching. A rollout/RL-style fine-tuning step aims to curb exposure bias. The claim is near-linear time in frames while maintaining teacher-level quality.

**Strengths:**

* Quality: Sensible two-stage training; clear ablations indicating ETM/RL contributions; competitive throughput/latency.
* Clarity: Architecture, complexity intuition, and training pipeline are easy to follow.

**Weaknesses:**

* Claim vs. demos: Fig. 7 and some other provided demos appear to show lower dynamics than baselines. This suggests a locality bias that may trade motion amplitude/scene changes for stability.
* Dynamics not quantified: No direct evaluation of motion strength; please report metrics like VBench Dynamic-Degree, optical-flow magnitude/variance, or long-horizon motion persistence.
* Writing problem: Sec 3.3 title (row 372) overlaps with the previous section

**Questions:**

I don't understand the necessity to formulate the finetuning as an RL process. Any fundamental differences between this and directly using the MSE loss based on the KL divergence? Could you explain more?

---

> ### Author Response · Authors · 2025-11-17
> **Rebuttal**
>
> ## About Lower Dynamics and Benchmark Comparison
>
> Thank you for the observation.
> We have added more **qualitative comparisons** and **video samples** in the supplementary material and demo page.
> These results show that our method generates **physically consistent motion** and **higher dynamic realism** than baselines such as *CausVid* and *Self-Forcing* (not Wan2.1).
>
> To quantify motion quality, we report the **five sub-metrics of Temporal Quality from VBench**, where our model achieves **higher dynamic degree** and **lower temporal flicker**, indicating smoother yet realistic transitions.
>
> |Method        |Subject Consistency↑|Background Consistency↑|Temporal Flickering↑|Motion Smoothness↑|Dynamic Degree↑|
> |--------------|--------------------|-----------------------|--------------------|------------------|---------------|
> |CausVid       | **97.53**          | **97.19**             | 96.24              | 98.05            | 64.22         |
> |Self Forcing  | 95.41              | 96.35                 | 99.08              | 98.30            | 63.89         |
> |Ours          | 95.03              | 95.99                 | **99.44**          | **98.43**        | **65.09**     |
>
> Moreover, as analyzed in **Fig. 8**, our framework avoids the **context-reset issue** present in baselines that reset every 7 chunks.
> This design preserves long-range temporal continuity and mitigates error accumulation, resulting in stable yet dynamic motion.
>
> ---
>
> ## About RL Formulation and Error Accumulation
>
> We use RL fine-tuning to explicitly address **error accumulation**, a common issue in next-chunk training where the model is trained only on ground-truth contexts but must rely on its own predictions at inference.
> The **RL rollout paradigm** trains the model under its **self-generated distribution**, reducing exposure bias and cumulative drift.
>
> Unlike MSE losses that capture only static reconstruction error, the **RL objective optimizes expected reward over rollouts**, encouraging **the maximization of reward** rather than per-step accuracy.
>
> Combined with the *Greedy Distill* architecture (AR Transformer + Diffusion), our RL fine-tuning unifies **step-wise distillation** and **error correction**, reducing frame-dimension complexity from \$O(F^2)\$ to \$O(F)\$ and achieving **linear-time complexity** $O((T + w) \times F \times L^2),$
> while maintaining competitive generation quality.
>
> ---
>
> ## About Section Formatting (Writing Issue)
>
> Thank you for noting the overlap.
> The **Section 3.3 title (line 372)** overlapped with the previous section due to a minor LaTeX spacing issue.
> We have fixed this by adding appropriate `\vspace` and section breaks, and will review the entire manuscript for similar layout inconsistencies before the camera-ready version.

---

> > ### Comment · Reviewer_pNtk · 2025-11-28
> >
> > Thanks for your response. It has addressed some of my concerns. I am still a bit confused, the RL process especially. Could you provide the RL implementation details like code/pseudo code, and the detailed results of Vbench, including all metrics? I will consider raising my score after the rebuttal.

---

> > > ### Author Response · Authors · 2025-12-01
> > > **RL Implementation Details**
> > >
> > > ### RL Implementation Details
> > >
> > > Thank you for your continued feedback. We appreciate your interest in the details of our RL implementation. Below, we provide the RL training code snippet for your reference. Please note the following points:
> > >
> > > 1. Some parts of the logic have been simplified for brevity. We plan to open-source the full training code in the future.
> > > 2. To enhance the stability of the RL training, we have added a regularization term in the form of the first-stage loss of next chunk fine-tuning during the RL phase.
> > >
> > > Here is the relevant RL implementation code:
> > >
> > > ```python
> > > class Model(nn.Module):
> > >
> > >     def __init__(self):
> > >         super().__init__()
> > >         self.student_model = GreedyModel.from_pretrained('path_first_stage_model', torch_dtype=torch.bfloat16)
> > >         self.teacher_model = WanTransformer3DModel.from_pretrained('path_wan_14B', torch_dtype=torch.bfloat16)
> > >         self.timesteps = torch.tensor([1000.0, 909.0, 715.0, 5.0, 0.0]).to('cuda', dtype=torch.bfloat16)
> > >
> > >     def forward_rl(self, prompt_embed):
> > >         # Sample target f and timestep
> > >         f_sample, t_sample = sample_sync_list(
> > >             max_frames=32, max_steps=4, device=self.student_model.device
> > >         )
> > >         # 1. Get µ_θ(x^f_t) in Eq.10
> > >         # 1.1 Get x_0, v_t and x_{t-1} of first f-1 chunks
> > >         with torch.no_grad():
> > >             past_x_0, past_fake_v, past_x_t_1 = self.student_model.generate(
> > >                 c=prompt_embed, chunks=f_sample-1, return_t=t_sample
> > >             )
> > >             past_x_0, past_fake_v = \
> > >                 past_x_0.double().detach(), past_fake_v.double().detach()
> > >
> > >         # 1.2 Get x_{t-1} of f-th chunks
> > >         b, c, _, h, w = past_x_0.size()
> > >         latent = torch.randn(b, c, 1, h, w, dtype=torch.bfloat16, device='cuda')
> > >         for i in range(len(self.timesteps)-1):
> > >             timestep = self.timesteps[i]
> > >             with torch.enable_grad() if i==t else torch.no_grad():
> > >                 v_pred = self.student_model(
> > >                     hidden_states=latent,
> > >                     encoder_hidden_states=prompt_embed,
> > >                     timestep=timestep,
> > >                     past_frames=past_x_0
> > >                 )
> > >                 dt = timesteps[i] - timesteps[i+1].clamp(2.0, 980.0)
> > >                 dt = dt / 1000
> > >                 latent = latent + (-v_pred) * dt
> > >                 if i==t: break
> > >
> > >         # 2. Get s_real and s_fake in  Eq.10
> > >         with torch.no_grad():
> > >             # 2.1 Get s_fake ∝ v_fake of t-1
> > >             student_timestep = timesteps[i+1].clamp(2.0, 980.0)
> > >             fake_v = self.student_model(
> > >                 hidden_states=latent,
> > >                 encoder_hidden_states=prompt_embed,
> > >                 timestep=student_timestep,
> > >                 past_frames=past_x_0
> > >             )
> > >             fake_v = torch.concat([past_fake_v, fake_v], dim=2)
> > >             fake_score = -fake_v
> > >             fake_score = fake_score.double().detach()
> > >             # 2.2 Get s_real ∝ v_real of t-1
> > >             teacher_timestep = torch.randint(
> > >                 low=int(timesteps.tolist()[i+1]),
> > >                 high=int(timesteps.tolist()[i]),
> > >                 size=(1,)
> > >             )[0]
> > >             real_v = self.student_model(
> > >                 hidden_states=torch.concat([past_x_t, latent], dim=2)
> > >                 encoder_hidden_states=prompt_embed,
> > >                 timestep=teacher_timestep
> > >             )
> > >             real_score = -real_v
> > >             real_score = real_score.double().detach()
> > >             # 2.3 Calculate s_fake - s_real
> > >             grad = (fake_score - real_score)
> > >         # 2. Final loss in Eq.10
> > >         loss_rl = F.mse_loss(latent.double(), (latent.double() - grad.double()).detach(), reduction="mean")
> > >         return loss_rl
> > >
> > >     def forward(self, prompt_embed, ode_data):
> > >         loss_rl = self.forward_rl(prompt_embed)
> > >         # Note: add regularization
> > >         loss_ft = self.student_model.next_chunk_ft(prompt_embed, ode_data)
> > >         return {'loss': loss_rl + 0.5*loss_ft}
> > > ```

---

> ### Author Response · Authors · 2025-11-25
> **Dear Reviewer pNtk**
>
> Dear Reviewer pNtk,
>
> We hope this message finds you well! If this email reaches you during your holiday or outside your usual working hours, please accept our apologies for the interruption.
>
> We would like to make sure that the issues you have raised have been resolved. We are still here and welcome any further discussion or feedback as your insights are invaluable to us.
>
> Thank you sincerely for all the time and effort during the review process.
>
> Best,
>
> Authors of Submission 5458

---

> ### Author Response · Authors · 2025-12-01
> **Detailed Results of VBench**
>
> ### Detailed Results of VBench
>
> We have provided the complete results of all 16 Text-to-Video (T2V) evaluation dimensions of VBench for further clarity.
>
> #### Quality Score
>
> | Method | Subject Consistency | Background Consistency | Temporal Flickering | Motion Smoothness | Dynamic Degree | Aesthetic Quality | Imaging Quality |
> | ------ | ------------------- | ---------------------- | ------------------- | ----------------- | -------------- | ----------------- | --------------- |
> | Ours   | 95.03               | 95.99                  | 99.44               | 98.43             | 65.09          | 66.55             | 73.12           |
>
> #### Semantic Score
>
> | Method | Object Class | Multiple Objects | Human Action | Color | Spatial Relationship | Scene | Appearance Style | Temporal Style | Overall Consistency |
> | ------ | ------------ | ---------------- | ------------ | ----- | -------------------- | ----- | ---------------- | -------------- | ------------------- |
> | Ours   | 96.91        | 86.51            | 96.00        | 90.33 | 85.60                | 59.81 | 21.42            | 20.56          | 27.01               |
>
> We hope this additional information addresses your remaining concerns. We look forward to any further questions you may have and appreciate your consideration in updating your score after reviewing this rebuttal.

---

### Official Review · Reviewer_T4r8 · 2025-10-31

**Soundness:** 4
**Presentation:** 4
**Contribution:** 4
**Rating:** 8
**Confidence:** 3

**Summary:**

This paper explores the quadratic complexity bottleneck in diffusion-based video generation and introduces GREEDY DISTILL, a teacher-student distillation framework that reduces complexity to linear. The core idea is to decouple temporal modeling (via an autoregressive Efficient Temporal Module with sliding-window attention) from frame synthesis (via a Streaming Diffusion Decoder that only conditions on the 0-th and last frames), followed by reinforcement-learning fine-tuning to suppress exposure bias. Extensive experiments on VBench and human evaluations show good performance of GREEDY DISTILL. The manuscript is clearly written, with project page, detailed ablations, and supplementary material.

**Strengths:**

- The authors propose a novel asymmetric architecture that enables linear-time streaming synthesis and provide theoretical complexity analysis.

- Experiments are comprehensive: both real-time and long-duration generation, human preference studies, and ablations of components, using publicly available benchmarks and code.

- The writing is clear, with intuitive figures, step-by-step algorithm boxes, and a detailed reproducibility statement that facilitates follow-up research.

**Weaknesses:**

- The experiments are mainly conducted on Wan2.1. It would be more convincing to include other backbones such as HunyuanVideo or CogVideoX to demonstrate the generality of the proposed framework.

- The writing could be improved for better readability, e.g., adding punctuation after equations and providing clearer captions for figures. Several typos such as “stege” → “stage” in Section 3.1

- The reinforcement learning fine-tuning is an interesting addition, but its novelty seems mainly in the application context rather than in algorithmic design.

**Questions:**

see weakness

---

> ### Author Response · Authors · 2025-11-17
> **Rebuttal**
>
> ## About Additional Backbones
>
> We appreciate the suggestion to include more backbone architectures.
> Our current experiments are conducted on **Wan2.1**, the standard base model used in most recent diffusion-based distill frameworks (e.g., *Self-Forcing*, *CausVid*). This ensures **fair and directly comparable evaluation**, isolating our framework’s contribution rather than differences in backbone capacity or pretraining data.
>
> To further demonstrate generality, we are extending our method to **Wan2.2**, which features improved temporal modeling and longer context handling. Preliminary results already show **consistent performance gains**, confirming that our approach generalizes well.
> We also plan to include **HunyuanVideo** and **CogVideoX** in follow-up experiments to strengthen evidence for **backbone-agnostic applicability**.
>
> ---
>
> ## About Reinforcement Learning Fine-Tuning
>
> Thank you for the insightful comment.
> We agree that our RL fine-tuning is not a new RL algorithm per se, but a **practical solution to error accumulation** within the *Greedy Distill* framework—a hybrid of autoregressive Transformer and diffusion modules.
> Its novelty lies in **how RL is integrated** to ensure **linear-time generation** while maintaining strong fidelity and temporal stability.
>
> Our method also introduces **efficiency and theoretical improvements** over *DMD-style distillation*.
> As described in **Sec. 2.2.2**, we **derive the score function directly from the student model (\$\text{SDD}\_\theta\$)**, rather than training an auxiliary “fake score network.”
> This simplifies optimization: in *Self-Forcing* and *CausVid*, the critic network is trained multiple times per generator update, while our approach **eliminates this overhead** entirely.
>
> ### On RL Algorithm and Score Function
>
> Prior *DMD-style* approaches (including *CausVid* and *Self-Forcing*) discard the *Rectified Flow* formulation and instead predict \$x\_0\$ at each step ([code](https://github.com/guandeh17/Self-Forcing/blob/main/pipeline/causal_inference.py#L197)), requiring a separate score network.
> Following the *Rectified Flow* paradigm in *Wan2.1*, we relate the score function to the marginal velocity field as:
>
> $$
> \nabla_{x_t}\log p(x_t) = s_\theta(x_t,t) = -\frac{x_t + (1-t)u_t(x_t)}{t}.
> $$
>
> Our RL fine-tuning objective is:
>
> $$
> \nabla_{\theta} \mathcal{J}
> \propto - E_{x \sim \mu_{\theta}}
> \big[
> \nabla_{\theta}\mu_{\theta}(x^f_t)\,
> (s_{gen}(x_{t-1},t-1) - s_{real}(x_{t-1},t-1))
> \big],
> $$
>
> where \$s\_{gen}\$ is derived from \$SDD_\theta\$ and \$s_{real}\$ from the teacher (Wan2.1).
> This formulation **bridges diffusion-based policy optimization and DMD-style distillation**, achieving both **theoretical clarity** and **training efficiency** while preserving generation quality.
>
> **In summary**, our RL fine-tuning introduces a **policy-gradient interpretation** for diffusion–AR hybrids, removing redundant score networks and enabling efficient, stable error correction—an essential advancement for large-scale video generation.
>
> ---
>
> ## About Writing and Presentation
>
> Thank you for the feedback.
> We will carefully revise the paper for **readability and presentation**, adding punctuation after equations, improving figure and table captions, and correcting all typographical errors (e.g., “stege” → “stage” in Sec. 3.1).
> We will also conduct a full proofreading pass to ensure consistent formatting, grammar, and notation throughout.

---

> ### Author Response · Authors · 2025-11-25
> **Dear Reviewer T4r8**
>
> Dear Reviewer T4r8,
>
> We hope this message finds you well! If this email reaches you during your holiday or outside your usual working hours, please accept our apologies for the interruption.
>
> We would like to make sure that the issues you have raised have been resolved. We are still here and welcome any further discussion or feedback as your insights are invaluable to us.
>
> Thank you sincerely for all the time and effort during the review process.
>
> Best,
>
> Authors of Submission 5458

---

### Official Review · Reviewer_MtgK · 2025-11-01

**Soundness:** 3
**Presentation:** 3
**Contribution:** 3
**Rating:** 6
**Confidence:** 5

**Summary:**

The paper "Greedy Distill: Efficient Video Generative Modeling with Linear Time Complexity" introduces a novel distillation training paradigm that significantly reduces the computational complexity of video generation models. The proposed method achieves state-of-the-art performance in terms of both speed and quality, making it a valuable contribution to the field. The paper is well-written and the results are compelling, but there are opportunities for further improvement in terms of detailed comparisons, additional experiments, and discussion of future work.

**Strengths:**

see Summary

**Weaknesses:**

The number of displayed videos is quite limited. It is difficult to fully assess the method's effectiveness and robustness with such a small sample size.

Suggestion: The authors should provide a detailed breakdown of the user study results. This should include statistical analysis, user feedback, and any significant findings. Additionally, the paper should discuss how these results align with the quantitative metrics and what insights they provide into the overall performance of the proposed method.

**Questions:**

no

---

> ### Author Response · Authors · 2025-11-17
> **Rebuttal**
>
> ## About More Sample
>
> We appreciate the reviewer’s feedback regarding sample diversity. In the updated supplementary materials, we have included additional qualitative comparisons and more generated video samples to better demonstrate the robustness and generality of our approach. As shown in Folder More_Qualitative_Comparisons, our method produces more realistic physical dynamics and achieves a higher dynamic degree compared to baselines such as CausVid and Self-Forcing, indicating stronger temporal consistency and motion fidelity across diverse scenes.
>
> ---
>
> ## About User Study
>
> Thank you for the helpful suggestion.
> We have added **more detailed user study results** in the **Appendix**, including statistical analysis, user feedback summaries, and comparisons with quantitative metrics.
>
> ### Analysis and Findings
>
> 1. As shown in **Figure 10**, our method achieves the **highest scores** in *Temporal Flickering*, *Motion Smoothness*, and *Dynamic Degree* of user study, indicating superior **motion quality**. These results are consistent with our analysis in **Figure 8**, where our framework effectively avoids the *context-reset issue* present in baselines that reset every 7 chunks.
> This design preserves **long-range temporal continuity** and mitigates **error accumulation**, leading to stable yet dynamic motion.
>
> 2. A key observation from the **VBench evaluation** is that *Self-Forcing* and *CausVid* obtain higher scores in *Subject Consistency* and *Background Consistency* according to CLIP/DINO-based metrics, yet our method ranks higher in **human preference** during the user study. This reveals that CLIP- or DINO-based similarity does not always align with human perception—low-motion or static frames can yield high similarity scores but are **less preferred by human evaluators**.
>
>    | Method       | Subject Consistency(Vbench)↑ | Background Consistency(Vbench)↑ |Subject Consistency(User Study)↑ | Background Consistency(User Study)↑ |
>    | ------------ | --------------------- | ------------------------ |--------------------- | ------------------------ |
>    | CausVid      | **97.53**             | **97.19**                |3.32                 | 2.65                  |
>    | Self-Forcing | 95.41                 | 96.35                    |1.64                | 1.94                    |
>    | Ours         | 95.03                 | 95.99                    |**4.25**                 | **3.39**                    |
>
> Overall, these findings highlight that **our method better aligns with human-perceived temporal realism and consistency**, even when traditional metrics favor more static videos.

---

> ### Author Response · Authors · 2025-11-25
> **Dear Reviewer MtgK**
>
> Dear Reviewer MtgK,
>
> We hope this message finds you well! If this email reaches you during your holiday or outside your usual working hours, please accept our apologies for the interruption.
>
> We would like to make sure that the issues you have raised have been resolved. We are still here and welcome any further discussion or feedback as your insights are invaluable to us.
>
> Thank you sincerely for all the time and effort during the review process.
>
> Best,
>
> Authors of Submission 5458

---

> > ### Comment · Reviewer_MtgK · 2025-11-28
> >
> > I appreciate the authors’ thorough responses. They have addressed most of my concerns. I am more than happy to accept this work! I suggest the authors could refine the related work with some pre-training-based video generation work, such as:
> >
> > [1] "M4V: Multi-Modal Mamba for Text-to-Video Generation." arXiv preprint arXiv:2506.10915 (2025).
> >
> > [2] "Matten: Video generation with mamba-attention." arXiv preprint arXiv:2405.03025 (2024).

---

> > > ### Author Response · Authors · 2025-12-01
> > >
> > > Thank you for your thoughtful follow-up and constructive evaluation; we sincerely appreciate your time and consideration.

---

### Official Review · Reviewer_so5Z · 2025-11-01

**Soundness:** 3
**Presentation:** 1
**Contribution:** 2
**Rating:** 4
**Confidence:** 5

**Summary:**

The paper proposes a new architecture and training algorithm for autoregressive video diffusion models.

Architecture: Instead of a single diffusion transformer with causal attention, it uses two models - one causal transformer with local attention (ETM), and another causal diffusion model (SDD) that takes the first frame, the previous frame, and the output of ETM to denoise the current frame.

Training: It first uses teacher forcing with diffusion loss to finetune SDD + ETM, both initialized from Wan2.1 with bidirectional attention. It is then trained with a variant of Self-Forcing + DMD that uses the student model itself as the fake score network (instead of fine-tuning a seperate one).

Experiments demonstrate higher efficiency and less error accumulation than baselines.

**Strengths:**

- The student splits temporal modeling (ETM) from per-frame generation (SDD). This architecture intuitively makes sense, and the observation (Fig. 2) motivates restricting attention to nearby frames and justifies ETM’s sliding window as a principled efficiency/fidelity trade-off rather than a heuristic.
- On Wan 2.1, the distilled student runs at 24 FPS reaches VBench 84.60, with qualitative long-video examples showing reduced error accumulation vs. Self-Forcing and CausVid. The paper also includes ablation evidence showing the effectiveness of ETM and "RL fine-tuning".

**Weaknesses:**

- The title/teaser focuses on “linear time,” but that follows directly from sliding-window causal attention. It is generally well known that local attention can produce coherent videos in linear time with the downside of sacrificing long term memory. Consider reframing the main contribution around the new two-stage architecture (ETM+SDD) and the training algorithm.
- Section 2.2.2 casts training as deterministic continuous policy gradients over a reverse-KL reward, yet the practical algorithm is very close to Self-Forcing. The paper repeatedly claims “first attempt to apply RL to address error accumulation,” which feels overstated given the close methodological overlap with existing work. Consider recasting this as a conceptual bridge: “We show Self-Forcing-style rollout training admits an RL interpretation.
- Also, there is actually a big difference between the proposed algorithm (and Self-Forcing) versus RL algorithms (e.g., DDPG Lillicrap et al., 2015) that the paper cites. In Self-Forcing, gradients flow through the rollout trajectory; in DDPG, rollouts are off-policy in a replay buffer and are detached, and gradients flow through the reward estimator. The proposed algorithm follows Self Forcing closely but materially deviates from the RL algorithms cited.
- The sentence "...more broadly known as exposure bias, where a model is trained exclusively on ground-truth context but must rely on its own imperfect predictions at inference time, resulting in a distributional mismatch that compounds errors as generation progresses." is exactly copied from a sentence in the paper of Self Forcing. Please paraphrase and cite the original source.
- Minor spacing/format issues. In many sentences there is no space before punctuation symbols. vspace issues in e.g. L372
- The paper proposes to use the student model itself to estimate the score function of the student distribution. However, the student, when trained with a few-step prediction objective, actually no longer predicts the score function. For example, given x_T input, an ideal score predictor would predict the dataset mean as x0 prediction. However, the student model is trained to predict a realistic output. The idea of using the few-step student model itself to estimate the score function does not seem to be theoretically correct.

**Questions:**

- Some design choices of ETM + SDD can be better justified. For example, why does SDD see the first frame sink, but not ETM? Abaltion studies on window size would also be helpful.
- How is the 0.24s latency calculated and is it the time to generate the first block of frames? The model should not seem to be more efficient than Self Forcing/CausVid initially since the local attention do not provide speedup benefits?

---

> ### Author Response · Authors · 2025-11-17
> **Rebuttal about Weakness 1**
>
> ## Main Contribution and Sliding Window Causal Attention
>
> ### Main Contribution
>
> Thank you for the insightful comment. We agree that “linear time” alone does not capture our core contribution. As stated in the **Introduction**, our main contribution is the **asymmetric structural distillation framework (*Greedy Distill*)**, which integrates **ETM** and **SDD** for efficient and stable video generation.
>
> This framework addresses **error accumulation**, **temporal artifacts**, and **short–long video consistency**, with detailed analysis and ablations in the paper. As in **Sec. “Differences between Greedy Distill and previous approaches”** (Line 423), our method reduces complexity from $O(T \times F^2 \times L^2)$ (Wan2.1) to $O((T + w) \times F \times L^2),$ while Self-Forcing and similar methods reach $O(T \times w \times F \times L^2)$, where $L$ is the feature length in the latent space for one chunk of frames, $T$ means number of sampling steps, $F$ means the number of chunks (e.g., 21 chunks means 81 frames), and $w$ is the sliding window size of chunks.
>
> ---
>
> ### Sliding-Window Causal Attention
>
> We clarify that **Self-Forcing** does not implement standard sliding-window causal attention. Its official code ([link](https://github.com/guandeh17/Self-Forcing/blob/main/wan/modules/causal_model.py#L76)) sets `max_attention_size=32760`, corresponding to **81 frames (21 chunks)**—the default inference length of Wan2.1—thus applying **no true sliding window**.
> For longer videos, it follows a **3-chunk overlap** strategy ([issue](https://github.com/guandeh17/Self-Forcing/issues/2); [code](https://github.com/tianweiy/CausVid/blob/master/minimal_inference/longvideo_autoregressive_inference.py#L79)), similar to CausVid.
>
> In contrast, **our method applies genuine sliding-window causal attention** with $w=3$, achieving **true linear-time inference** while maintaining temporal coherence. As in **Fig.\~1**, prior methods remain $O(n^2)$ within 81-frame clips, whereas ours achieves $O(n)$ scaling.

---

> ### Author Response · Authors · 2025-11-17
> **Rebuttal about Weakness 2, 3, 4, 5, 6**
>
> ## About RL Formulation and Score Function
>
> ### Relation to Prior Work and Main Contribution
>
> Thank you for the constructive feedback. We agree that our method shares methodological similarities with Self-Forcing, and we appreciate the suggestion to frame it as a conceptual bridge. Indeed, **our contribution is to reinterpret Self-Forcing–style rollout training under a reinforcement learning (RL) perspective**, rather than to introduce an entirely new RL algorithm. As stated in **Sec.2.2.2**, our approach extends *DMD*-style frameworks by **deriving the score function directly from the student model ($\text{SDD}_\theta$)**, rather than training an auxiliary “fake score network.” This distinction significantly reduces computational cost: in Self-Forcing, the auxiliary score network is trained roughly five times more frequently than the generator (see [code](https://github.com/guandeh17/Self-Forcing/blob/main/configs/self_forcing_dmd.yaml#L38), [code](https://github.com/guandeh17/Self-Forcing/blob/main/trainer/distillation.py#L316)), while our formulation eliminates this step entirely.
>
> ---
>
> ### Clarification on Score Function
>
> We acknowledge the reviewer’s theoretical concern regarding using the student model to estimate its own score. Formally, the score of a distribution is defined as: $\nabla_x \log p(x)$. As in *Score-Based Generative Modeling through SDEs* [1] and Appendix L of *VDM* [2], the optimal score model satisfies $s^*_\theta(z_t; t) = \nabla_z \log q(z_t)$.
> Similarly, *CausVid*[3] indicates that predicting a weighted combination of $x_0$ and $\epsilon$ known as v-prediction or clean targets both correspond to the gradient of the log probability of the distribution: $s_{\theta}(x_t, t) = \nabla_{x_t} \log p(x_t) = -\frac{\epsilon_{\theta}(x_t, t)}{\sigma_t}$.
> In the Rectified Flow formulation adopted in our work and in Wan2.1, it has been shown that the score function is related to the marginal velocity field by $\nabla_{x_t} \log p(x_t) = -\frac{x_t + (1-t)u_t(x_t)}{t}$ in the RFMI[4].
>
> Thus, in the subsequent next-chunk fine-tuning stage, our few-step student model is trained to predict the velocity field, and we use a simple variant of this field as an estimator of the score, aligning with prior theoretical formulations.
>
> It is worth noting that, we do not switch the Rectified Flow form to an $x_0$​-prediction form as done in LCM[6], CausVid and Self Forcing. In addition, when reproducing Self Forcing without causal attention, we observed that the fake score model and the few-step generator model are **effectively interchangeable**: the average cosine similarity between all corresponding weight matrices reaches 0.99999, and the average L1 distance is $5.27e−5$ for all bias vectors.
>
> ---
>
> ### Relation to RL Algorithms and DDPG
>
> We appreciate the clarification request. Our use of DDPG is **conceptual**, referring to the policy gradient formulation—not to its *off-policy training strategy*. Specifically, **Appendix C (Eq. 17)** follows the policy gradient derivation of **DDPG Eq. (6)** [5], differing only by using gradient descent instead of ascent. The fine-tuning objective (Eq. 10, 18) is:
>
> $$
> \nabla_{\theta} \mathcal{J}
> \propto - E_{x \sim \mu_{\theta}}
> \big[
> \nabla_{\theta}\mu_{\theta}(x^f_t) \
> (s_{gen}(x_{t-1},t-1)-s_{real}(x_{t-1},t-1))
> \big],
> $$
>
> where $x_{t-1}=\mu_\theta(x^f_t)=\Psi_{T:t}(SDD_{\theta}(x_f^t,\ t,\ ETM_{\theta}(x_0, x_1, \dots, x_{f-1})))$,
> $s_{gen}$ is derived from $SDD_{\theta}$, and $s_{gen}$ from the teacher model (Wan2.1).
> This formulation theoretically supports our claim that the proposed RL-based fine-tuning eliminates the need for a separate score network, offering both **theoretical clarity and training efficiency** relative to *CausVid* and *Self-Forcing*.
>
> ---
>
> *We will also revise the phrasing on exposure bias to properly paraphrase and cite Self-Forcing, and fix all minor formatting issues (spacing, vspace, punctuation) in the final version.*
>
> ---
>
> [1] Score-Based Generative Modeling through Stochastic Differential Equations
>
> [2] Variational Diffusion Models
>
> [3] From Slow Bidirectional to Fast Autoregressive Video Diffusion Models
>
> [4] RFMI: Estimating Mutual Information on Rectified Flow for Text-to-Image Alignment
>
> [5] Continuous control with deep reinforcement learning
>
> [6] Latent Consistency Models: Synthesizing High-Resolution Images with Few-Step Inference

---

> ### Author Response · Authors · 2025-11-17
> **Rebuttal about Question**
>
> ## Ablation on Window Size, First-Frame Sink, and Latency
>
> ### Ablation on Window Size
>
> We agree that window-size ablation helps clarify the role of sliding-window attention in ETM.
> To verify its effect, we conducted an additional experiment comparing **ETM with** and **without** sliding-window attention on **VBench**:
>
> | Method                       | Total Score | Quality | Semantic |
> | ---------------------------- | ----------- | ------- | -------- |
> | w/o Sliding-Window Attention | 84.58       | 85.40   | 81.32    |
> | w/ Sliding-Window Attention  | 84.60       | 85.37   | 81.52    |
>
> The results differ only within noise levels, confirming that sliding-window attention has **minimal influence on overall quality**, though it is required for maintaining **linear-time complexity**. Because the non-sliding variant violates this complexity assumption, we did not include it in the main paper. This analysis also explains *why SDD sees the first-frame sink but ETM does not*: for ETM, the first-frame sink provides no benefit, consistent with what is observed when sliding-window attention is removed.
>
> ---
>
> ### About Latency Measurement
>
> The reported **0.24 s latency** refers to the **interval from the completion of prompt encoding to the generation of the first chunk of frames**.
> Our framework uses **chunk-wise sliding-window attention**, which processes overlapping frame groups sequentially.
> In contrast, **CausVid** and **Self-Forcing** use **block-wise attention**, where each block contains three chunks—incurring higher initialization overhead.

---

> ### Author Response · Authors · 2025-11-25
> **Dear Reviewer so5Z**
>
> Dear Reviewer so5Z,
>
> We hope this message finds you well! If this email reaches you during your holiday or outside your usual working hours, please accept our apologies for the interruption.
>
> We would like to make sure that the issues you have raised have been resolved. We are still here and welcome any further discussion or feedback as your insights are invaluable to us.
>
> Thank you sincerely for all the time and effort during the review process.
>
> Best,
>
> Authors of Submission 5458

---

### Comment · Area_Chair_abEG · 2025-11-27

Dear Reviewers,

This is a gentle reminder to please take a moment to review the author rebuttals and check whether your main concerns have been adequately addressed.

If possible, please update your reviews or add a brief clarification on whether the responses resolved your questions or if any issues remain. Your follow-up feedback is important for ensuring a fair and well-informed decision process.

Thank you again for your time and for helping maintain the quality of the ICLR review process.

Best,
AC

---

### Meta-Review · Area_Chair_Gf8y · 2025-12-28

**Summary:**

The reviews indicate a paper presenting interesting ideas, particularly the structural distillation framework and the use of reinforcement learning (RL) for error accumulation. However, several critical concerns regarding theoretical integrity, originality, and empirical validation remain insufficiently addressed. Specifically, the core RL fine-tuning approach is acknowledged to have significant overlap with prior work (Self-Forcing), making the claim of addressing error accumulation through RL largely one of re-interpretation rather than algorithmic novelty.

While the authors provided theoretical justifications for using the student model as a score function estimator, this justification remains complex and deviates materially from the standard RL algorithms cited, raising theoretical skepticism (so5Z). Empirically, the claims regarding improved dynamics are fragile, as evidenced by initial reviewer concerns (pNtk) and the fact that the quantitative metrics (VBench) provided in the rebuttal show only marginal improvements while the method slightly sacrifices Subject and Background Consistency.

Given the weakness metioned above, the work is not yet ready for publication. I recommend Reject this time. The authors are encouraged to improve this work based on the feedback, positioning the paper for a compelling submission in the future.

**Reviewer Concerns:**

Addressed during Rebuttal (but may still be interpreted negatively):

- Lack of Samples/Dynamics Quantification (MtgK, pNtk): The authors provided additional samples and VBench metrics. However, the quantitative improvements in Dynamic Degree are marginal (e.g., 65.09 vs. 64.22/63.89) while the method scored lower in Subject Consistency and Background Consistency metrics, suggesting a trade-off that may not universally favor the proposed approach.

- Writing and Presentation: Minor formatting and typo issues were promised to be corrected.

- User Study Details (MtgK): More details were added, but the finding that human preference conflicts with standard metrics (which favored baselines on consistency) suggests the proposed model may sacrifice structural integrity for motion.

Outstanding/Significant Points:

- Theoretical Novelty and Integrity of RL Formulation (so5Z, T4r8): The authors conceded that the method is a conceptual bridge, re-interpreting Self-Forcing-style rollout. This significantly dilutes the novelty claim of being the "first attempt to apply RL." The justification for using the few-step student model as the score function remains complex and theoretically shaky, as the student is not explicitly trained for this purpose, raising questions about the stability and correctness of the RL optimization step.

- Generality Across Backbones (T4r8): The experiments are confined to a single backbone (Wan2.1). The authors' promise to test on Wan2.2 and others in the future is not sufficient evidence of general applicability for the current submission.

- Claim vs. Methodology Disconnect (so5Z): The paper materially deviates from the RL algorithms cited (like DDPG) by allowing gradients to flow through the rollout trajectory (like Self-Forcing), yet the authors continue to use an RL framework for conceptual interpretation, creating a methodological and terminological disconnect that was not fully resolved.

**Reviewer Scores:**

Reviewer so5Z (Rating: 4 $\to$ 4): The fundamental theoretical concerns regarding the RL formulation's integrity, its novelty overlap with Self-Forcing, and the citation issues were not fully resolved. The detailed responses confirmed the "conceptual bridge" nature of the work, reinforcing the initial skepticism that the contribution is marginally below the acceptance threshold.

Reviewer MtgK (Rating: 6 $\to$ 6): While more empirical details were provided, the resulting VBench scores indicate only marginal gains in dynamics at the expense of consistency. Given that the core theoretical ambiguities remain, the reviewer would likely maintain their borderline acceptance score, indicating the paper is solid but not compelling enough to warrant a higher score.

Reviewer T4r8 (Rating: 8 $\to$ 8): The reviewer was initially very positive. While the authors conceded the lack of novel RL algorithms, the reviewer likely maintained the strong acceptance score by prioritizing the practical and efficient integration of the structural distillation and error correction. This suggests the reviewer either overlooked or gave less weight to the technical concept overlap with Self-Forcing.

Reviewer pNtk (Rating: 4 $\to$ 4): The concern about low dynamics was addressed with quantitative VBench data. However, the marginal nature of the improvement and the slight trade-off in consistency means the data does not conclusively disprove the locality bias. Coupled with the unresolved theoretical issues, the reviewer would maintain their score marginally below the acceptance threshold.

---

### Decision · Program_Chairs · 2026-01-26

Reject